

# Metabarcoding dietary analysis of coral dwelling predatory fish demonstrates the minor contribution of coral mutualists to their highly partitioned, generalist diet

Matthieu Leray[1,2,3], Christopher P. Meyer[3] and Suzanne C. Mills[1,2]

[1] USR 3278 CRIOBE CNRS-EPHE-UPVD, CBETM de l'Université de Perpignan, Perpignan Cedex, France
[2] Laboratoire d'Excellence "CORAIL"
[3] Department of Invertebrate Zoology, National Museum of Natural History, Smithsonian Institution, Washington, D.C., USA

## ABSTRACT

Understanding the role of predators in food webs can be challenging in highly diverse predator/prey systems composed of small cryptic species. DNA based dietary analysis can supplement predator removal experiments and provide high resolution for prey identification. Here we use a metabarcoding approach to provide initial insights into the diet and functional role of coral-dwelling predatory fish feeding on small invertebrates. Fish were collected in Moorea (French Polynesia) where the BIOCODE project has generated DNA barcodes for numerous coral associated invertebrate species. Pyrosequencing data revealed a total of 292 Operational Taxonomic Units (OTU) in the gut contents of the arc-eye hawkfish (*Paracirrhites arcatus*), the flame hawkfish (*Neocirrhites armatus*) and the coral croucher (*Caracanthus maculatus*). One hundred forty-nine (51%) of them had species-level matches in reference libraries (>98% similarity) while 76 additional OTUs (26%) could be identified to higher taxonomic levels. Decapods that have a mutualistic relationship with *Pocillopora* and are typically dominant among coral branches, represent a minor contribution of the predators' diets. Instead, predators mainly consumed transient species including pelagic taxa such as copepods, chaetognaths and siphonophores suggesting non random feeding behavior. We also identified prey species known to have direct negative interactions with stony corals, such as *Hapalocarcinus* sp, a gall crab considered a coral parasite, as well as species of vermetid snails known for their deleterious effects on coral growth. *Pocillopora* DNA accounted for 20.8% and 20.1% of total number of sequences in the guts of the flame hawkfish and coral croucher but it was not detected in the guts of the arc-eye hawkfish. Comparison of diets among the three fishes demonstrates remarkable partitioning with nearly 80% of prey items consumed by only one predator. Overall, the taxonomic resolution provided by the metabarcoding approach highlights a highly complex interaction web and demonstrates that levels of trophic partitioning among coral reef fishes have likely been underestimated. Therefore, we strongly encourage further empirical approaches to dietary studies prior to making assumptions of trophic equivalency in food web reconstruction.

Corresponding author
Matthieu Leray,
leray.upmc@gmail.com

## INTRODUCTION

Anthropogenic stressors are impacting all ecosystems on Earth, causing both drastic changes in the structure of communities and a reduction in biodiversity (*Wright, 2005*; *Hoegh-Guldberg & Bruno, 2010*). Predators are among the most vulnerable trophic group, and have long been known to play a crucial role in stabilizing ecosystems by generating top-down forces and trophic cascades (*Paine, 1966*; *Paine, 1969*). Yet, because all predator species are not functionally equivalent, understanding how species partition their diet as well as their ecological role in food webs have become a major focus to help predict the consequences of their decline on ecosystem services (*Harley, 2011*).

A detailed knowledge of a predator's diet is a key element for deciphering its ecological function. Among the numerous techniques used in the literature to characterize a predator's diet, PCR-based molecular analysis of gut contents is among the most powerful because species-diagnostic DNA fragments can be detected even after several hours of digestion (*Symondson, 2002*). Moreover, the availability of versatile PCR primers targeting short hypervariable DNA regions combined with a high-throughput sequencing platform now offer the possibility to characterize the dietary breadth of any predator (*Pompanon et al., 2012*; *Leray et al., 2013a*). The ecological influence of a predator may then be inferred from its dietary selectivity as well as the traits and functional role of prey consumed (*Chapman et al., 2013*). On land, this tool is already proving invaluable for understanding the biological control potential of insect predators (*Mollot et al., 2014*) and the ecological effects of large herbivores (*Kowalczyk et al., 2011*) and carnivores (*Shehzad et al., 2012*). The use of high-throughput sequencing for understanding trophic links in marine systems has been more limited to date (*Leray et al., 2013a*).

On coral reefs, one of the most diverse and threatened of ecosystems, predatory fish feeding on benthic invertebrates are the dominant trophic category. They often dwell within the reef framework where they feed upon diverse communities of small cryptic species that are known to perform a variety of functions including direct positive or negative interactions with stony corals, the foundation species of the coral reef ecosystem (reviewed by *Stella et al., 2011*). Some invertebrate taxa promote the survival and growth of corals by slowing the progression of coral diseases (*Pollock et al., 2012*), protecting corals against corallivores (*Glynn, 1980*; *Glynn, 1983*; *McKeon & Moore, 2014*; *Rouzé et al., 2014*),removing sediments from their coral host (*Stewart et al., 2006*; *Stier et al., 2012*) and alleviating detrimental effects of coral competitors or parasites (*Stier et al., 2010*). Other invertebrates have deleterious effects on corals as they are known vectors of coral diseases (*Sussman et al., 2003*; *Williams & Miller, 2005*), are parasites of stony corals (*Humes, 1985*; *Shima, Osenberg & Stier, 2010*) or feed upon coral polyps (*Turner, 1994*; *Rotjan & Lewis, 2008*; *Rawlinson et al., 2011*) sometimes causing extensive and widespread coral mortality (*Leray et al., 2012a*; *Kayal et al., 2012*). As a consequence, the feeding behavior of these

predatory fish may have significant cascading effects on the dynamics of stony corals and subsequently the dynamics of the whole coral reef ecosystem, but it has proven challenging to understand their ecological role.

The flame hawkfish (*Neocirrhites armatus*), arc-eye hawkfish (*Paracirrhites arcatus*) and coral croucher (*Caracanthus maculatus*) are common predatory fish species on Indo-Pacific coral reefs. They co-occur among the branches of Pocilloporids (genus *Stylophora* and *Pocillopora*), one of the most important reef building corals, along with a wide diversity of invertebrates (*Patton, 1974*; *Coles, 1980*; *Odinetz, 1983*; *Stella, Jones & Pratchett, 2010*). These invertebrates include both coral mutualistic (family: Trapeziidae and some Alpheidae) and parasitic (family: Cryptochiridae) decapod species (*Simon-Blecher & Achituv, 1997*), which are potential prey for coral dwelling fish. A field manipulation of the two *Pocilloporid* obligate species, the flame hawkfish and the coral croucher (habitat specialists), highlighted that their presence among the branches of *Pocillopora eydouxi* reduced total abundance and diversity of decapod recruits by 34% and 20% respectively (*Stier & Leray, 2014*). These predators modified the composition and abundance of key mutualists (coral crabs, genus: *Trapezia*), whose benefits to *Pocillopora* are known to be both density- and diversity-dependent (*Stier et al., 2012*). Predator removal experiments have also shown that the presence of arc-eye hawkfish decreases the density of coral associated mutualist damselfish (*Holbrook, Schmitt & Brooks, 2011*). Preliminary molecular dietary analysis using traditional cloning showed the presence of coral mutualists in the gut contents of both hawkfish species (*Leray et al., 2013b*), but sampling and sequencing effort were too limited to understand their contribution to each species' diets.

In the present study, we use a high throughput sequencing approach targeting the mitochondrial Cytochrome c. Oxidase subunit I gene (COI) (also referred to as metabarcoding approach, *Taberlet et al., 2012*) to describe the dietary breadth of these predators. The study was conducted in Moorea, French Polynesia, where an extensive library of COI DNA barcodes, including all *Pocillopora* associated species, has been built by the BIOCODE project (*Leray et al., 2012b*). Implications of each predator's feeding behavior are further discussed in light of our findings.

## METHODS

### Predator and prey collections

Twenty-five adult specimens of each of the three predator fish species were speared after sunset, which corresponds to peak feeding time for all three species (M Leray, pers. obs., 2010), in the lagoon of the North shore of Moorea on the 8th, 10th and 15th of July 2010. We limited our collections to a single site (17°28′40S; 149°50′25W, Fig. 1) where coral populations had been little impacted by the recent outbreak of the corallivorous seastar, *Acanthaster planci* (*Adjeroud et al., 2009*; *Kayal et al., 2011*; *Rouzé et al., 2015*). Adults of the flame hawkfish and coral croucher always co-occurred among *Pocillopora* branches, whereas adult arc-eye hawkfish were occasionally present. Fish were individually preserved in cold 50% ethanol *in situ* after which their digestive track was dissected within 3 h and preserved in eppendorf tubes containing 80% ethanol.

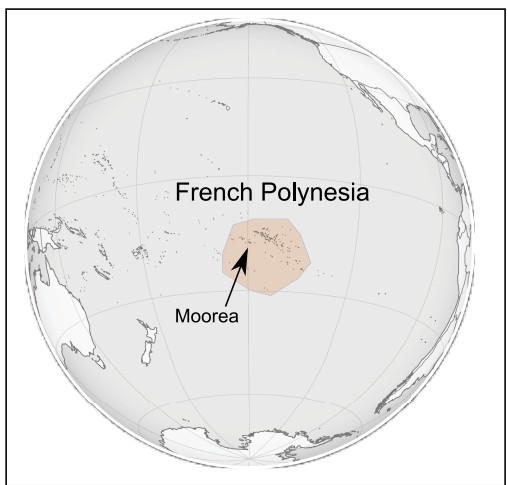
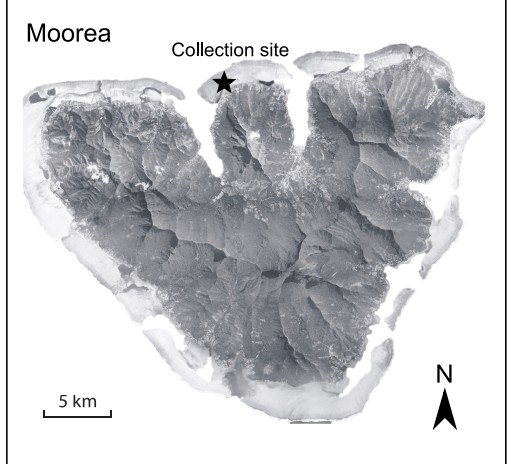

**Figure 1** Map of the study location.

## Laboratory protocol

The total content of the digestive track of each fish was dissected and used for total genomic DNA extraction using the QIAGEN DNeasy Blood & Tissue kit. Genomic DNA was then purified using the PowerClean DNA clean-up kit (MO BIO) to remove potential PCR inhibitors. We used a single set of versatile PCR primers (mlCOIintF/jgHCO2198, *Geller et al., 2013*; *Leray et al., 2013a*) known to perform well across the diversity of marine invertebrates, to amplify a 313bp region of the mitochondrial Cytochrome c. Oxidase subunit I (COI) region from each gut content sample. Moreover, this primer set was recently shown to provide reliable estimates of relative abundance for metabarcoding benthic samples (*Leray & Knowlton, 2015*). Because predator DNA co-amplification is known to impede prey detection (*Vestheim & Jarman, 2008*), predator-specific annealing blocking primers were included at ten times the concentration of versatile primers during PCR reactions as in *Leray et al. (2013a)*. All primer sequences are provided in Table 1. The PCR cocktail and touchdown temperature profile used in this study can both be found in *Leray et al. (2013a)*. Three PCR replications per sample were generated, pooled, gel excised to ensure that all primer dimers were screened away, purified using QIAGEN® MinElute columns and eluted in 12 µl of elution buffer. PCR product concentration was measured with the Qubit® Fluorometer (Invitrogen, Carslsbad, California, USA).

We pooled equimolar amounts of the combined amplicons per individual gut content for each predator species (e.g., 25 flame hawkfish gut content samples were pooled together) and 500 ng of PCR product was used per species for library preparation for Roche 454 FLX sequencing. Amplicons were end-repaired and dA-tailed using the NEBNext Quick DNA Sample Prep Reagent Set 2 chemistry (New England BioLabs, Ipswitch, Massachusetts, USA). We then performed a ligation of 454 Multiplex Identifiers (a total of three, each one containing a recognizable sequence tag) using the FLX Titanium Rapid Library MID Adaptors Kit (Roche, Basel Switzerland). Finally, the ligated PCR product of each sample was purified using Agencourt AMPure beads (Beckman Coulter

**Table 1  List of primers used in this study.**

| Primer label | Sequence (5′–3′) | Reference |
|---|---|---|
| mlCOIintF | GGWACWGGWTGAACWGTWTAYCCYCC | (*Leray et al., 2013a*) |
| jgHCO2198 | TAIACYTCIGGRTGICCRAARAAYCA | (*Geller et al., 2013*) |
| Narmatus_Blocking | CAAAGAATCAAAACAGGTGTTGATAAAGA-C3 | (*Leray et al., 2013b*) |
| Parcatus_Blocking | CAAAGAATCAGAACAGATGTTGGTAAAGA-C3 | (*Leray et al., 2013b*) |
| Cmaculatus_Blocking | CAAAGAATCAGAATAGGTGTTGGTACAGA-C3 | Herein |

Genomics, Danvers, Massachusetts, USA), eluted in 40 µl of TE buffer, and the three samples pooled together for sequencing at the Duke Institute for Genome Sciences and Policy (Duke University, North Carolina, USA). Note that the three samples of the present study were combined with several other samples in the same 454 run.

## Analysis of FLX sequencing data

We followed a sequence data analysis pipeline optimized for analyzing large COI datasets. The pipeline detailed in *Leray et al. (2013a)* takes advantage of the coding properties of the barcoding region to discard all dubious sequences.

The initial step denoised flowgrams using Pyronoise (*Quince et al., 2011*) implemented in Mothur (*Schloss et al., 2009*). We then further quality filtered the dataset by removing any reads that met the following criteria: shorter than 200bp; more than two mismatches in the primers sequence; any ambiguous base calls (e.g., "N"); or with any homopolymer regions longer than 8bp. Remaining sequences were subsequently aligned to a high quality reference dataset (Moorea BIOCODE barcode library) based on amino acid translations using the option "enrichAlignment" in MACSE (*Ranwez et al., 2011*) and all sequences with any of the following were also discarded: stop codon; frame shift; insertion; or more than three deletions. Finally, potential chimeric sequences identified using UCHIME (*Edgar et al., 2011*) were removed to obtain a high quality sequence dataset for downstream analysis.

To evaluate prey richness and composition, sequences were clustered in Operational Taxonomic Units (OTUs) using a Bayesian algorithm implemented in CROP (*Hao, Jiang & Chen, 2011*). This program delineates OTUs based on the natural distribution of sequence dissimilarity in the data and within a range of sequence similarity values defined by the user. This approach performs better for clustering sequences obtained from environmental samples than a fixed dissimilarity cutoff (e.g., 5%) because they contain a diversity of phyla that differ in their rate of COI evolution. The lower and upper bound variance were set to 3 and 4 respectively (which corresponds to 6% and 8%) as they were shown to provide the best results for marine invertebrates (*Leray et al., 2013a*; *Leray et al., 2013b*). Following OTU delineation, a representative sequence per OTU was used for taxonomic identification using BLAST searches in the local BIOCODE database and in GENBANK. We considered that there was a species level match when sequence similarity was at least 98% (*Machida et al., 2009*; *Plaisance et al., 2009*). Whenever sequence similarity was lower

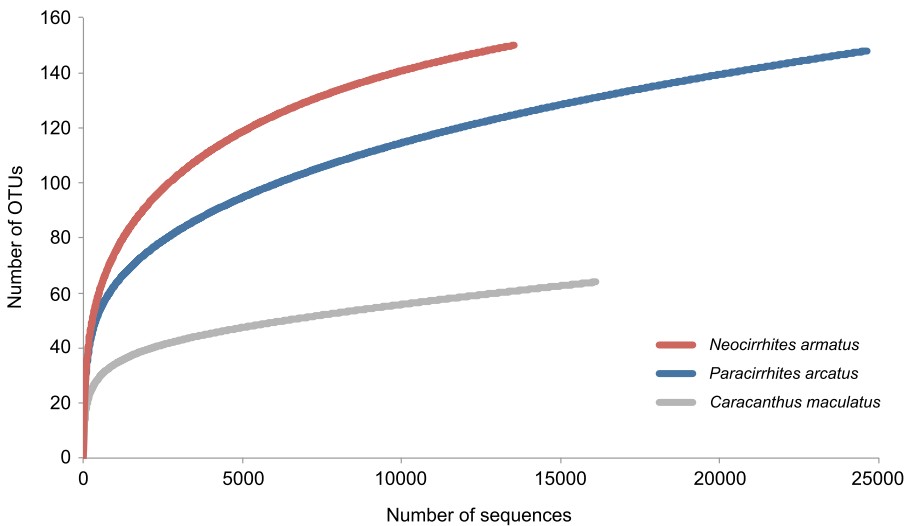

**Figure 2** Rarefaction curves to evaluate the completeness of the sequencing effort at describing the diversity of dietary items in the gut contents of three coral reef fish species.

than 98%, we used a Bayesian approach implemented in the Statistical Assignment Package (SAP, *Munch et al., 2008*) to assign OTUs to a higher taxonomic group. SAP conducts assignments by building 10,000 unrooted phylogenetic trees from a collection of homologue sequences retrieved from a sequence database. It then calculates the probability that a query sequence belongs to a monophyletic group within that set of homologues. Here, we allowed SAP to retrieve 50 homologues from GENBANK with >70% sequence similarity to each query sequence (i.e., each OTU representative sequence) and accepted taxonomic assignments at an 80% posterior probability cutoff. Importantly, SAP can only assign sequences to taxonomic groups that are represented in the database, as is also the case with other assignment methods. Therefore, to minimize misidentification at lower taxonomic levels, we only report assignments to the phylum, class and order levels (Appendix S1).

## RESULTS

We obtained 69,663 reads of which 54,283 high quality reads were retained for downstream analysis (arc-eye hawkfish: 24,629; flame hawkfish: 13,536; coral croucher: 16,118). The Bayesian clustering algorithm delineated 292 OTUs in the gut contents of the three predatory fish species (Appendix S1). The number of dietary items was much lower in the gut contents of the coral croucher (64 taxa) than in both arc-eye (147 taxa) and flame hawkfish (149 taxa). BLAST searches provided high-resolution taxonomic assignments (>98% similarity) for 149 OTUs (51%) (Appendix S1) and the statistical assignment approach enabled the identification of 76 additional OTUs to a higher taxonomic level (26%). 67 OTUs (22.9%) remained unidentified (labeled as "Unidentified" in Appendix S1). None of the rarefaction curves reached a plateau (Fig. 2) which indicates that further sequencing effort would be necessary for a more exhaustive dietary analysis of these predatory fish.

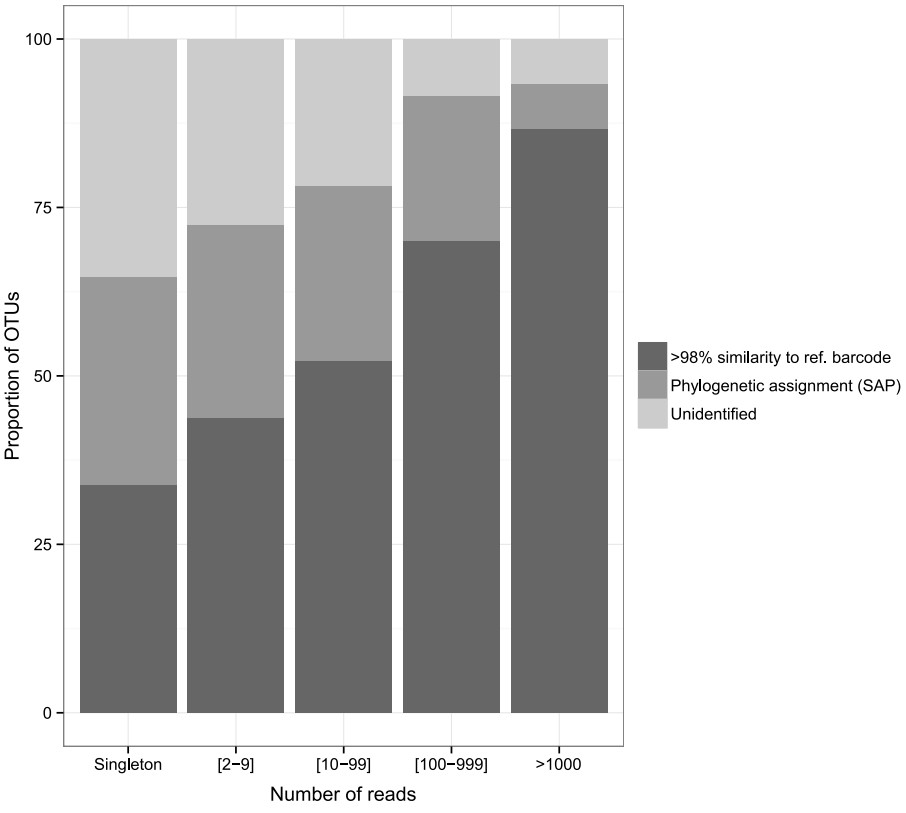

**Figure 3 Proportion of identified OTUs in relation to the number of sequences they represent.** Whenever OTU sequence similarity to a reference barcode was <98%, we used the Phylogenetic Bayesian assignment tool implemented in SAP to assign OTUs to a higher taxonomic group.

The diversity of dietary items spanned 25 classes belonging to 17 phyla. Malacostraca was the dominant taxonomic prey group (36.7%, 21.5% and 43.7% for the arc-eye hawkfish, the flame hawkfish and the coral croucher, respectively). The arc-eye hawkfish also consumed numerous species of Actinopterygii (17.7% total OTUs) and Maxillopoda (10.9% total OTUs). A significant proportion of the flame hawkfish and coral croucher's diet was represented by Maxillopoda (12.1% and 6.2% total OTUs, respectively) and Gastropoda (9.4% and 4.7% total OTUs, respectively). Eighteen OTUs (28%) detected in the gut contents of the coral croucher remained unidentified. Direct matches to reference barcodes (>98% similarity) were more prevalent among Actinopterygii (94.1%), Malacostraca (74.1%) and Gastropoda (79.2%) compared to Maxillopoda (40%). Moreover, direct matches were more prevalent for OTUs represented by large numbers of sequences (Fig. 3). Almost nine of ten OTUs (86.7%) matched reference barcodes if they were common in the amplicon libraries (>1,000 sequences), whereas only a third (33.8%) of the single sequences matched a reference sequence. Probability of a match increased as the number of sequences increased (1: 33.8%; [2–9]: 43.7%; [10–99]: 52.2%; [100–999]: 70%; >1,000: 86.7%; Fig. 3).

Most Malacostraca OTUs were decapods (81.5%, 46.9% and 78.6% for the arc-eye hawkfish, the flame hawkfish and the coral croucher respectively— Appendix S1). All

three predatory fish fed upon *Pocillopora* obligate decapod species, but they represent a minor fraction of the total diversity of the prey they consumed (arc-eye hawkfish: 2%; flame hawkfish: 4%, coral croucher: 9.3%). Among them, we detected five coral crab species of the genus *Trapezia* that are mutualists of *Pocillopora* (*Trapezia bidentata*, *T. serenei*, *T. rufopunctata*, *T. areolata* and *T*. spp). These mutualists also represented a minor proportion of sequences in the gut contents of the arc-eye and flame hawkfish (proportion of total sequences: 5.6% and 2.4%; proportion of decapod sequences: 9.1% and 12.7%, respectively). *Pocillopora* mutualists represented a higher proportion of the coral croucher's diet with 15.3% of the total number of sequences and 47.9% of the total number of decapod sequences.

Additional trophic links involving non-decapod prey are of particular interest for understanding the effect of predators on coral and its associated communities. Predatory fish had fed upon coral associated planktivorous damselfishes of the family Pomacentridae (*Dascyllus flavicaudus*: 0.02%, 0% and 0.12%, *Chromis viridis*: 0.01%, 0.69% and 0% of total sequences in the diet of the arc-eye hawkfish, the flame hawkfish and the coral croucher, respectively) that benefit the growth of the coral host (*Holbrook et al., 2008*). Interestingly, Anthozoa were represented by two OTUs among which the host *Pocillopora* itself accounted for 20.8% and 20.1% of total number of sequences in the guts of flame hawkfish and coral croucher, but was completely absent from the gut of the arc-eye hawkfish. On the other hand, *Hapalocarcinus sp*, a gall crab considered a coral parasite, was recovered in the diet of both the arc-eye and flame hawkfish. Both hawkfish had also consumed vermetid snails known for their deleterious effects on coral growth (*Shima, Osenberg & Stier, 2010*). *Harpiliopsis beaupresii*, a caridean shrimp associated with *Pocillopora* but whose function is unknown, was also detected in the gut contents of the coral croucher. Almost 10 percent (8.3%) of the coral croucher's diet is composed of two snails (*Drupa ricinus* and *Pascula muricata*). Finally, predators had also consumed pelagic taxa including members of Maxillipoda, Chaetognatha and Hydrozoa (Appendix S1).

Prey species were remarkably partitioned among predators (Fig. 4). Almost eighty percent (79.5%) of prey species had been consumed by only one predator species (232 of 292). Eighteen percent ($N = 52$) were found in two predator diets and only eight prey species (>3%) had been ingested by all three predatory fish species analyzed. Of the shared components, the arc-eye hawkfish and the coral croucher had consumed 14 taxa in common among which six were Malacostraca. The arc-eye and flame hawkfish shared 29 prey taxa with a majority of Actinopterygii and Malacostraca. Prey sharing was lowest (nine OTUs; of which six were Malacostraca) between the two species that were always found co-occuring together in the coral host, the flame hawkfish and coral croucher. Analyses that included only prey OTUs consisting of >1% of either of the three species diets according to the relative abundance of reads demonstrate even greater partitioning (Fig. 4). Only six of the sixty-six prey items were shared at a proportion greater than 1% in any two fish species diets, and no prey species were shared among all three. Of the 66 prey items making up at least 1% of any diet, nine out of ten were consumed by only one predator.

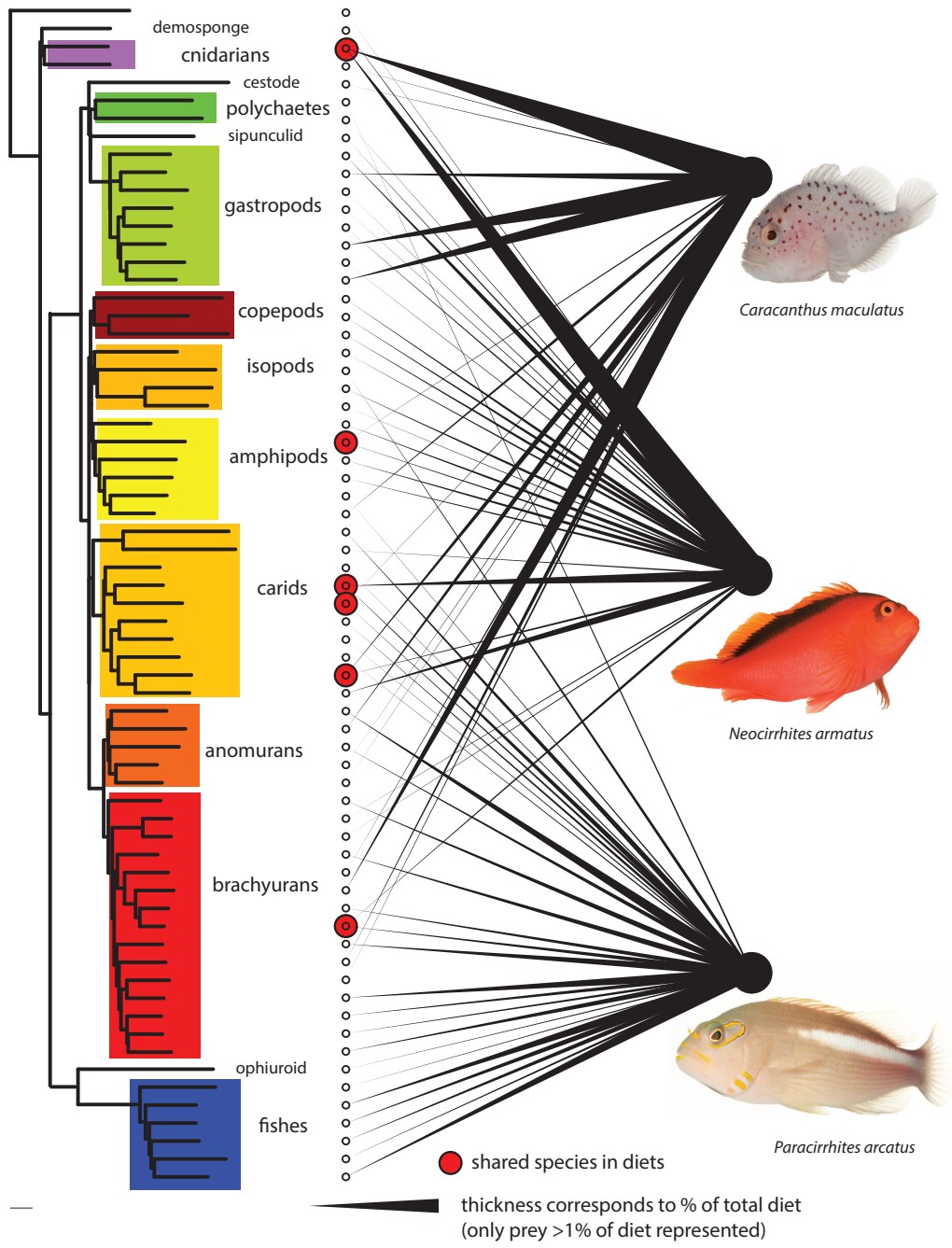

**Figure 4 Dietary partitioning among the three predatory fish species.** Left neighbor-joining phylogeny using LogDet distance model based on a constraint topology of major clades represents relationship among the 66 prey OTUs that comprise >1% of any one species diet. Thickness of linkages to right represents relative proportion of predatory diets. Six shared species are highlighted with circles. Fish images courtesy of D. Liittschwager. The 66 OTUs are highlighted in Appendix S1.

## DISCUSSION

Dietary analysis can be a powerful approach to gain insights into the ecological role of reef-dwelling predatory fish, but low taxonomic resolution in prey identification often obscures the complexity of trophic links (*Longenecker, 2007*). For example, the diet of the arc-eye hawkfish, flame hawkfish and coral croucher previously described from morphological identification of prey remains in gut contents was considered to be simply composed of small benthic crustaceans (class: Malacostraca) (*Bachet, Zysman & Lefevre, 2007*). Preliminary DNA analysis using traditional cloning revealed a breadth of prey species in the guts of the arc-eye and flame hawkfish, the majority of which were crustaceans 18 of 24 (75%) and 21 of 31 (68%) respectively (*Leray et al., 2013b*). This study highlights that a metabarcoding approach significantly increases the taxonomic scope by documenting an even broader taxonomic distribution of species consumed by hawkfishes (Appendix S1). The coral croucher diet also includes a wide spectrum of prey demonstrating that all three predatory fish feed broadly across community diversity. Our results highlight the importance of collecting empirical dietary data to understand processes of species coexistence in this high diversity marine ecosystem.

The ecological influence of a predator is contingent upon the prey it consumes. Their feeding behavior may induce cascading effects that will depend on the type of association that the prey they consume (or interfere with) have with keystone species. For example, in terrestrial ecosystems where up to 90% of flowering plant species use animal pollinators for reproduction (*Bushmann & Nabhan, 1996*), a predator's effect on plant reproductive success, growth and survival will depend on its relative consumption of pollinators and phytophageous insects (*Dukas, 2005*; *Knight et al., 2006*). Similarly, some coral reef dwelling predatory fish may either disrupt benefits to corals if they derive a significant proportion of their diet from coral mutualists or alternatively alleviate deleterious effects on corals if they consume coral parasites. Invertebrate communities occurring among the branches of live *Pocillopora* corals in Moorea or elsewhere in the Pacific are typically composed of a preponderance of decapod mutualists (>80% of diversity and abundance in live *Pocillopora*—see *Patton, 1974*; *Coles, 1980*; *Odinetz, 1983*; *Stella, Jones & Pratchett, 2010*; *Leray et al., 2012a*). Based on previous cloning studies (*Leray et al., 2013b*) only the arc-eye hawkfish consumed functionally important prey (*Trapezia tigrina*). With increased sequencing depth herein, we demonstrate that while many other mutualist decapod species do occur in the diets of the arc-eye hawkfish, flame hawkfish and coral croucher (5.6%, 2.4% and 15.3% of sequence abundance, respectively; Appendix S1), they represent a much smaller proportion of the diet than would be expected from their density in natural communities. Interestingly, we found evidence of the *Pocillopora* obligate pontoniid shrimp *Harpiliopsis beaupressi* but no detection of congeneric *H. depressa* and *H. spinigera* in the predators' gut contents, despite their very high abundance reported on head-size *Pocillopora* in Moorea (*Leray et al., 2012a*). It is also surprising not to discover *Alpheus lottini* in the diets of the three species although this is a common species found in all living *Pocillopora* observed and known to have beneficial effects on coral survivorship (*Stier et al., 2012*). Overall, our data indicate a non random pattern of prey consumption atypical of

an opportunistic feeding behavior (where prey would be consumed in proportion to their abundance—*Heinlein, Stier & Steele, 2010*) which suggests the outcome of coevolutionary dynamics between *Pocillopora* associated predator and prey.

Nevertheless, while our metabarcoding dietary analysis suggests limited predation pressure on mutualists, a four-month recruitment experiment conducted on the North shore of Moorea in 2009 showed a lower abundance of mutualists in corals where the coral croucher and the flame hawkfish occurred (*Stier & Leray, 2014*), a pattern that may be driven by non-consumptive effects of predators. For example, competent larvae may preferentially settle on corals where predators are absent. Regardless of the mechanisms, such predator effects have important implications for coral performance, because density and composition of mutualist assemblages are known to be important for the quality of the services provided to their host (*Stier et al., 2012*; *Rouzé et al., 2014*).

In addition, our metabarcoding analyses of gut contents revealed for the first time predation on a gall crab (*Hapalocarcinus sp*) and vermetid snails (genus: *Dendropoma*), which are considered detrimental to the coral host (*Simon-Blecher & Achituv, 1997*; *Shima, Osenberg & Stier, 2010*). Vermetid snails are particularly prevalent in Moorea where they can reduce coral growth by up to 81% and survival by up to 52% (*Shima, Osenberg & Stier, 2010*). Predation on parasites may compensate for the negative effects of the reduction in density of decapod mutualists in corals facing environmental stressors. We also recovered a significant proportion of sequences belonging to *Pocillopora* from the flame hawkfish and the coral croucher gut contents, which suggest that these predatory fish also feed on mucus released by their biogenic habitat. The absence of *Symbiodinium* COI sequences from our dataset also supports the consumption of mucus rather than coral polyps. Alternatively, *Pocillopora* DNA may have been sufficiently abundant and well preserved in the gut contents of mucus feeding prey (e.g., Trapeziidae) to be co-amplified (*Harwood et al., 2001*; *Sheppard et al., 2005*). Importantly though, *Pocillopora* was completely absent from the arc-eye hawkfish diet which also includes Trapeziid species, suggesting minimal secondary consumption or associated eDNA amplification. Overall, high-resolution dietary data are revealing a highly complex interaction web with very specialized functional roles played by each species. This highlights the shortcomings of the functional groups approach commonly used to evaluate redundancy and complementarity among coral reef species (*Naeem & Wright, 2003*; *Micheli & Halpern, 2005*).

Fine-scale spatial partitioning commonly occurs among coral reef fish species (*Robertson & Lassig, 1980*; *Waldner & Robertson, 1980*; *Ebersole, 1985*; *Bouchonnavaro, 1986*; *Munday, Jones & Caley, 1997*; *Depczynski & Bellwood, 2004*; *Gardiner & Jones, 2005*) but the extent of food partitioning remains controversial (*Longenecker, 2007*). In fact, most early work investigating differences in diet among reef fish species showed high levels of diet overlap (*Hiatt & Strasburg, 1960*; *Randall, 1967*; *Hobson, 1974*; *Talbot, Russell & Anderson, 1978*; *Harmelin-Vivien, 1979*; *Anderson et al., 1981*; *Bouchonnavaro, 1986*; *Ross, 1986*; *Depczynski & Bellwood, 2003*; *Kulbicki et al., 2005*; *Longenecker, 2007*; *Castellanos-Galindo & Giraldo, 2008*) which has led many to the conclusion that trophic partitioning was not a mechanism promoting species coexistence on coral reefs. However, these studies,

which rely on morphological identification of semi-digested prey remains in gut contents grouped food items into broad categories therefore impeding accurate measures of partitioning (*Longenecker, 2007*). Alternative strategies such as field observations of feeding behavior (*Pratchett, 2005*; *Pratchett, 2007*; *Pratchett & Berumen, 2008*) or a combination of gut content and stable isotope analyses (*Ho et al., 2007*; *Nagelkerken et al., 2009*) helped describe dietary differences between closely related species, but generalizations about the importance of trophic partitioning for the maintenance of coral reef diversity remain difficult. In the present study, high-resolution molecular data highlight an unexpected level of dietary partitioning among the three study species. While both hawkfish species are from the same family (Cirrhitidae), they share only a single prey item at greater than 1% of either of their diets (*Trapezia serenei*). There is also a minor dietary overlap between the coral croucher (family Caracanthidae) and the flame hawkfish that were always found co-occurring in *Pocillopora* and are known to rarely venture outside the branching structure provided by their host coral (*Hiatt & Strasburg, 1960*; *Stier & Leray, 2014*). These results demonstrate that levels of trophic partitioning have likely been underestimated. We strongly encourage further empirical approaches to dietary studies prior to making assumptions of trophic equivalency in food web reconstruction (*Leibold & McPeek, 2006*).

The extent to which secondary prey co-amplification could lead to errors in food web analysis has not been quantified in marine systems (see *Sheppard et al., 2005* for an example in a terrestrial system). In the present dataset, numerous prey species identified in fish gut contents are either grazers or detritivores (e.g., isopods, amphipods, decapods, ophiuroids and gastropods) and are therefore unlikely to consume each other. Some fish species detected in the gut contents are higher-level predators (e.g., *Caranx melampygus*) that could consume benthic grazers and detritivores as adults, but they were most likely fed upon at a younger developmental stage (egg, larva or juvenile) given the size of predators. Demospongiae, Ascidiacea and Gymnolaemata represented by few or a single sequence in the dataset were, however, possibly ingested unintentionally as secondary prey or epiphytes on the carapace of spider crabs (e.g., *Menaethius monoceros* and *Perinia tumida*). Parasites of prey (e.g., parasitic isopods of coral crabs of the genus *Trapezia*, Appendix S1) and parasites of a predator's digestive track (e.g., Trematoda and Cestoda) may also confound food web reconstructions and care should be taken to consider the targeted roles these fish predators have on various parasites. The recovery of secondary prey may artificially inflate dietary partitioning if those lower levels are also partitioned. However, we expect the amount of DNA that these secondary prey items represent in the guts of our target predators should be minor and highly digested in comparison to primary prey. A recent metabarcoding analysis of benthic samples (*Leray & Knowlton, 2015*) showed evidence of a correlation between amount of DNA and number of reads. Thus if secondary prey is quickly degraded, those taxa should be represented by one or few reads only. The present dataset shows minor dietary overlap both with and without rare OTUs (<1% of total OTUs, Fig. 4), further supporting our conclusions regarding the extent of trophic partitioning among all three fish species.

Importantly, our analysis shows that in-depth sequencing would enable a more comprehensive representation of trophic links in this multi-faceted ecosystem. Additional reads would provide more OTUs matching reference barcodes (in GENBANK, BOLD or BIOCODE) but also a higher proportion of unidentified OTUs represented by a single sequence ("singleton", Fig. 3) that are likely to be either (1) small taxa underrepresented in DNA barcode libraries (*Leray et al., 2013a*), or (2) the product of sequencing artifacts despite our very stringent quality filtering based on amino-acid translation. Further barcoding initiatives aiming to catalogue small life forms (e.g., meiofauna) will be crucial to advance our understanding of food webs. Systematic removal of singletons may also be used as a conservative measure, although most of them likely represent valid taxa (*Huse et al., 2010*). As coral reef ecosystems decline worldwide, understanding the role of predator species in a dominant, yet largely understudied trophic category, is essential. Our study highlights the tremendous potential of metabarcoding as an approach to provide unprecedented taxonomic resolution in the diet of coral dwelling predatory fish. We encourage that further work should be conducted to understand the ecological role of reef dwelling fish and invertebrates.

## ACKNOWLEDGEMENTS

We thank Gustav Paulay, Arthur Anker, Joseph Poupin and the BIOCODE teams who collected both marine and terrestrial specimens, the "Centre de Recherche Insulaire et Observatoire de l'Environnement (CRIOBE) de Moorea" and the Richard B. Gump field station in Moorea for logistical support.

### Funding

Funding was provided by the Gordon and Betty Moore Foundation, France American Cultural Exchange program (FACE)/Partner University Fund (PUF), the Smithsonian Institution fellowship program and the Agence National de Recherche, ANR-11-JSV7-012-01 Live and Let Die. The funders had no role in study design, data collection and analysis, decision to publish, or preparation of the manuscript.

### Grant Disclosures

The following grant information was disclosed by the authors:
Gordon and Betty Moore Foundation.
France American Cultural Exchange program (FACE)/Partner University Fund (PUF).
Smithsonian Institution fellowship program and the Agence National de Recherche: ANR-11-JSV7-012-01.

### Competing Interests

Matthieu Leray is a postdoctoral fellow at the National Museum of Natural History, Smithsonian Institution. Christopher P Meyer is a research zoologist at the Department of Invertebrate Zoology, National Museum of Natural History, Smithsonian Institution.

## Author Contributions

- Matthieu Leray conceived and designed the experiments, performed the experiments, analyzed the data, contributed reagents/materials/analysis tools, wrote the paper, prepared figures and/or tables, reviewed drafts of the paper.
- Christopher P. Meyer conceived and designed the experiments, contributed reagents/materials/analysis tools, prepared figures and/or tables, reviewed drafts of the paper.
- Suzanne C. Mills conceived and designed the experiments, contributed reagents/materials/analysis tools, reviewed drafts of the paper.

## Animal Ethics

The following information was supplied relating to ethical approvals (i.e., approving body and any reference numbers):

Approval was granted from our institutional animal ethics committee, le Centre National de la Recherche Scientifique (CNRS), for sacrificing and subsequently dissecting fish (Permit Number: 006725). None of the fish species are on the endangered species list and no specific authorization was required from the French Polynesian government for collection.

## DNA Deposition

The following information was supplied regarding the deposition of DNA sequences:

An alignment of all OTU representative sequences is provided in Appendix S2 and the denoised sequence dataset is deposited in the Dryad Repository DOI 10.5061/dryad.v0p71).

## Supplemental Information

Supplemental information for this article can be found online at http://dx.doi.org/10.7717/peerj.1047#supplemental-information.

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
