# Peer review of "Metabarcoding dietary analysis of coral dwelling predatory fish demonstrates the minor contribution of coral mutualists to their highly partitioned, generalist diet"

_PeerJ, doi:10.7717/peerj.1047_

## Round 0.1 · original submission · Minor Revisions

Both reviewers provide helpful feedback and you should address their comments carefully, In particular, regarding potential amplicon biases, prey analysis, sequence depth of 454 reads.

Reviewer 1 ·

Basic reporting

Some of your sentences are unclear when discussing the percentages of reads assigned to a species or OTU. You use ‘respectively’ frequently but not always with a clear distinction of the respective terms. For example, line 149 uses “respectively” but it is unclear to me what these taxonomic classes are respective to. I think you have a great introduction that exemplifies how your work contributes to a broad scale trophic question. However, I think the introduction and discussion would benefit from more references regarding trophic cascades and broad ecological concepts, perhaps in other systems. Lastly, you have an entire section in your discussion section that references a figure that is not uploaded with the manuscript. This to me is a major revision. I cannot comment well on this section of your discussion without seeing figure three.

Experimental design

You have a very general experimental design. I think the investigation did not have as high of technical standard as it could have but that it is still a contribution. For example, using the 454FLX platform is fine for the long read lengths but in general requires more reads than what you obtained. Use of an Illumina platform might benefit this study. With the longer reads offered by Illumina, a 312bp amplicon can easily be assessed. I think the experimental design is novel and still supports use of advanced sequencing technologies to report increased diversity in a range of ecosystems. This is more of a recommendation for future work. Also, more details regarding your data analyses and bioinformatics pipeline would be nice to see. See line references in the "general comments" section of this review.

Validity of the findings

Again, I think more details about the use of SAP to assigned OTUs with less than 98% would provide more support for the findings. That none of your rarefaction curves plateaued should probably be discussed more. Do you think more reads would increase the number of OTUs that were assigned by BIOCODE and GenBank or would more reads reveal even more prey diversity? Did you try any basic statistics to look for correlations between the prey species? I recommend trying some basic statistics such as a PCA to see of the few prey OTUs(species) found in all three species which was explaining the most variation. Additionally, you could see if there is statistical significance between the mutualist species the fish consumed versus the parasitic. Lastly, glancing at your alignment revealed some significantly shorter sequences. I would like to know the ranges of the length of the sequences analyzed. Were all sequences analyzed longer than 300bp?

Again, I think your results are valid and a great contribution to the scientific community. How predator fishes impact the coral reef ecosystem is very relevant to maintaining biodiversity. Clearly analyzing gut contents using advanced sequencing is yielding novel relationships we were previously unable to detect. I think in general more details regarding your data analyses would enhance this paper.

Additional comments

I think this is an interesting study and method to assess coral reef ecosystems to ask broad scale ecological questions. However, in general I think this paper needs more details, especially in the methods and how the data were analyzed before being ready for submission. The results and discussion do a great job at discussing only what the results are capable of showing. You do not make unnecessary speculations. There are many formatting inconsistencies in your references and in the in-text citations that need to be addressed. Please go back and review all commas after “et al.,”, check for periods after “i.e.”, be consistent in how you spell out mitochondrial cytochrome c oxidase subunit, either spell out sentences that start with a number or revise sentence structure so it does not start with a number and thoroughly mine your references to remove inconsistencies.

I think figure two could provide within clade variation since this is discussed so much in the paper. Perhaps this would need to be an additional figure that shows within clade assignments rather than only the clades. I think this is relevant given the format of your discussion.

Most importantly you need to provide figure three!

I also might recommend adding a location figure, perhaps a broad scale subset map of Moorea with a larger zoomed in map of the sampling location.

I would also like to know more details about overall reef health in the area you caught the fish and how that compares to other reefs in the area. Then future studies might look for abundance trends in these predator species that could correlate to reef health.

Again, this was a very enjoyable paper to read and provides a novel approach to assess a key ecological interaction on reefs. I suggest major revisions until reviewing figure three.


Line Number References for formatting consistencies and grammar corrections
Line 36: Add comma after citation “Stella et al., 2011”
Line 57: There is an extra space after 34 before the % sign
Line 59: Change ‘know’ to ‘known’
Line 65-66: You don’t capitalize ‘mitochondiral cytochrome c oxidase subunit I gene” here but you do on lines 92-92. Be consistent.
Line 66: Add comma after citation
Line 89: This sentence reads poorly with the use of ‘individual columns’. Consider revising sentence structure.
Line 91: You are missing a parenthesis to close primers and author names appropriately.
Line101: Spell out EB buffer since it is your first use.
Line 107: How many MIDs did you use?
Line 109: Do you mean “Agencourt AMPure beads”?

Line 134: At what point did you cut-off similarity for your added statistical approach? Did you assess sequence similarity down to 50%? 95% for the 76 additional OTUs? You just say when sequences had less than 98% similarity you used SAP. If they are too dissimilar, even assigning an OTU to a higher taxonomic group may be risky. An OTU sequence that is very different from a reference sequence should probably be removed from the dataset.

Appendix I. If there is neither a BIOCODE or GenBank number but the OTU has a ‘class’ and ‘lowest taxon’, how was it assigned? SAP? This is not clear in the Appendix and additional text should be added to the caption.

Line 149: This sentence is a little confusing as it reads. Do you mean the dietary items spanned across 18, 16 and 11 taxonomic classes for the three fish species respectively? If not, then why not say the dietary items spanned across the total number of taxonomic classes you found?

Line 158: This sentence really makes me question how you assigned your OTUs with less than 98% similarity to a BIOCODE or GenBank reference sequence. Clearly reads that are most abundant in your dataset will have a higher probability of being a true read (i.e. a true representative sequence found in your study). Fewer reads mean it could be a sequence of a species in your mixed community OR more likely it is sequencing error. Consider providing more details regarding your quality checks in your analyses section within ‘methods’ and perhaps more details about your Bayesian approach to assign OTUs to a higher taxonomic group.

Line 159: Do you have a citation you could add here to show why you feel confident assigning an OTU with only one sequence? Were these 33.8% matching with 98% similarity? It would be nice to know how similar sequences were to the reference sequence OTUs that had less than 100 reads.

Line 160: Is a single sequence enough to confirm that sequence is a real sequence and not sequencing error?
Line 165: This line reads confusing since you use ‘respectively’. Is the reader to assume it is the same order of the three predator species from the previous sentence? Which species consumed 2% Pocillopora obligate decapod species? 4%? 9.3?
Line 171: I had to read this sentence a few times to understand what these percentages were representing. Consider removing “respectively “ and changing sentence structure. Perhaps something like “mutualist sequences representing 15.3% and 47.9% of the total decapod sequences.”
Line 186: Your figure reference seems out of place. Is “Fig. 2” only referencing that the predators consumed pelagic taxa? Figure two shows the extremely diverse diet of these predatory fish. I would reference Figure 2 sooner in this paragraph.
Line 187: Where is figure 3? I only have two to review!
Line 188: Which predator?
Line 193: Do you mean six of the OTUs shared between the flame hawkfish and coral croucher were Malacostraca? If so, reword to say (nine OTUs: of which six were Malacostraca).
Line 194: Make include past tense “Analyses that included…”
Lines 194-199: What do you mean by prey item? Is this the same as prey sequences? Species? OTUs?
Line 200: Did you not trim reads? (Looking at your Appendix two it appears some of your reads are short.)
Line 227: Need commas after et al. (This happens many times throughout your paper. You need to go back and check all your citation formatting for these inconsistencies.)
Line 231: You are missing a % sign
Line 239: Add a comma after “Heinlein et al., 2010”
Line 266: i.e. is missing a period before Trematoda
Line 413: I believe Vibrio shiloi should be italicized

Discussion lines 268-277: This is confusing given your results paragraph line 187-201. In your results you state the two hawkfish share 29 taxa and the arc-eye and coral croucher share 14 taxa. You do not list the number of taxa in the nine OTUs shared between the coral croucher and flame hawkfish, which would be nice to know. In the discussion you say there is more dietary overlap between the coral croucher and arc-eye than between the arc-eye and flame hawkfish but not according to the number of taxa. Consider defining ‘dietary overlap’ in the discussion.

Need to go through all your references. You are inconsistent in how you capitalize the second names of journals, for example you use “Coral Reefs” in line 322 but use “Coral reefs” on line 346. You have quite a few formatting errors in your references. Also correct “Molecular Ecology” versus “Molecular ecology”, “BMC bioinformatics” or “BMC Bioinformatics”, etc.

Figure 1: A minor edit but I would prefer the sentence to end in “…gut contents of three fish species.”

If an average 454FLX Titanium run yields around 700,000 did you pool other samples with this to explain your low coverage? You have few reads in the realm of sequencing, making your conclusions somewhat suspect to making some of the comparisons you do.

In general, don’t start sentences with numbers, such as 33.8% on line 159. This should be spelled out if it is starting a sentence or don’t make this the start of the sentence. Reword the sentence to put something before 33.8%.

Reviewer 2 ·

Basic reporting

See "general comments to authors"

Experimental design

See "general comments to authors"

Validity of the findings

See "general comments to authors"

Additional comments

Leray, Meyer and Mills studied the feeding preferences of three coral dwelling predatory fish. They used a metabarcoding approach using the 454 next generation sequencing platform. Overall, their study, manuscript and data, is sound and provides new insights into the feeding ecology of these fish. Although I read with great interest the paper, I found that they overlooked some methodological considerations. For instance, given the method based on shot-gun sequencing, there is little mention on secondary acquisition of OTUs, already eaten by their prey, particularly given that some of the prey were other fish (Actinopterygii). They only mention Trapezia as a by-catch but the values of other taxa, such as crustaceans, could be inflated by secondary acquisition. Do they have any idea in which extent? They also oversee the fact that some of the prey could be in the form of eggs and larvae and not only adults (for instance coral mucus could be larvae, eggs or tissue, and pomacentrids could be their fertilized eggs). Also, it is clear that any barcoding approach has a sampling bias to some groups, such as vertebrates. How does barcoding previous sampling affect their results? I did not find any mention about it and the amount of unidentified prey is very high in the study (23%). The authors sell quite well the importance of understanding predation in food webs as an important process leading to cascading effects and feedbacks. However, the discussion has little direct links of they results with a new understanding of predation and the mutualistic relationships with corals, or coral reef trophic ecology in general; at least it is not explicit. Likewise, I was expecting a more clear interpretation for the differences among the three fish species with more discussion on niche differentiation vs. competition. I suggest that a reviewed version of this manuscript should address those points in more detail. Below there are some minor comments on the manuscript.


Line 18-19. Please provide literature support, particularly since the previous sentences targets anthropogenic stressors.

Line 75. Reference supporting the peak feeding time for these fish.

Line 76. Why did your limit your sampling to one site?

Line 80. Why cold 50% ethanol and not a higher concentration?

Line 81. Why not 96% ethanol for better DNA preservation?

Line 104. Change “i.e.” to “e.g.”, which is what you mean.

Line 128. Please review use and the meaning of i.e.= id est (that is) and e.g. = exempli gratia (for example), throughout the manuscript actually.

Line 201. Update.

Line 240. How exactly your data support this statement? I do not really see it.

Line 253-255. Is this a byproduct or are you suggesting this is part of the mutualistic relationship with the coral?

Line 257. How do you know is mucus? It could be larvae or adult tissue? You did not you check for Symbiodinium OTUs, or did you?

---

## Round 0.2 · accepted · Accept

Congratulations on a nice contribution.